# Peripheral Nervous System Adverse Events after the Administration of mRNA Vaccines: A Systematic Review and Meta-Analysis of Large-Scale Studies

**DOI:** 10.3390/vaccines10122174

**Published:** 2022-12-17

**Authors:** Yu-Hsin Lai, Hong-Yu Chen, Hsin-Hui Chiu, Yi-No Kang, Shi-Bing Wong

**Affiliations:** 1Department of Pediatrics, Taipei Tzu Chi Hospital, Tzu Chi Medical Foundation, New Taipei 231405, Taiwan; 2Department of General Medicine, Taipei Tzu Chi Hospital, Tzu Chi Medical Foundation, New Taipei 231405, Taiwan; 3School of Medicine, College of Medicine, Taipei Medical University, Taipei 11031, Taiwan; 4Cochrane Taiwan, Taipei Medical University, Taipei 11031, Taiwan; 5School of Medicine, Tzu Chi University, Hualien 97004, Taiwan

**Keywords:** COVID-19, BNT162b2, mRNA-1273, Bell’s palsy, Guillain–Barré syndrome

## Abstract

Although neurological complications after the administration of vaccines against coronavirus disease 2019 (COVID-19) are rare, they might result in long-term morbidity. This study was designed to determine the risk of peripheral nervous system (PNS) adverse events after the administration of mRNA vaccines against COVID-19. Large-scale randomized controlled trials (RCTs) and cohort studies were systematically searched in databases, and 15 cohort studies were included in the synthesis. Among all PNS adverse events, only Bell’s palsy and Guillain–Barré syndrome (GBS) had sufficient data and were included for further analysis. Individuals who received mRNA vaccines had a higher risk of Bell’s palsy than the unvaccinated group, and the risk of Bell’s palsy after BNT162b2 was significantly higher than after mRNA-1273. Regarding GBS, no significant difference in the risk was observed between BNT162b2 and the unvaccinated group, but BNT126b2 introduced a higher risk of post-vaccinated GBS than mRNA-1273. In conclusion, PNS adverse events, especially Bell’s palsy, should be carefully observed after mRNA vaccination against COVID-19. With the opportunity of vaccination campaigns on such a large scale, further investigation and surveillance of post-vaccination neurological adverse events should also be established.

## 1. Introduction

The coronavirus disease 2019 (COVID-19) has infected more than 400 million individuals and caused more than 5.5 million deaths worldwide [1]. Various vaccines have been developed against the severe acute respiratory syndrome coronavirus 2 (SARS-CoV-2) and have been confirmed to be safe and effective tools for preventing severe COVID-19 [2]. Among these vaccines, mRNA vaccines against COVID-19, including BNT162b2 and mRNA-1273, have proven to be highly effective in reducing the risk of COVID-19 by significantly reducing COVID-19-associated hospitalizations and deaths [3,4,5,6], and were used in more than 100 countries worldwide. BNT162b2 is the first vaccine against COVID-19 which received full FDA authorization [7]. In addition, BNT162b2 and mRNA-1273 are the only vaccines that can be administered to persons younger than 12 years of age in the United states [8]. Therefore, BNT162b2 and mRNA-1273 are highly recommended to prevent the morbidity and mortality of SARS-CoV-2 infection [8]. However, based on a meta-analysis, the global vaccine acceptability was approximately 73%, and the acceptability was even lower in healthcare workers who were more vulnerable to COVID-19 infection [9]. The main reason for the unwillingness of individuals to receive COVID-19 vaccines was vaccine-associated side effects [9], highlighting the importance of public health messaging about vaccine safety. The proper recognition of adverse events after vaccination is mandatory to improve vaccine acceptance.

Although neurological complications after the administration of vaccines against COVID-19 are rare, they might result in long-term morbidity [10]. Peripheral nervous system (PNS) adverse events have been observed after the extensive use of BNT162b2 and mRNA-1273 vaccines globally, including Bell’s palsy [11] and Guillain–Barré syndrome (GBS) [12,13,14]. Theoretically, mRNA vaccines can induce innate immune activation and interferon production from a combined effect of mRNA and lipids, which may transiently break peripheral nerves [15]. However, the real-world data of Bell’s palsy and other PNS adverse events after the administration of mRNA vaccines are controversial [11,16,17]. In this meta-analysis, we collected and examined large-scale cohort studies recruiting more than 10,000 participants which potentially revealed post-vaccination adverse events because of the rarity of occurrence [18] to illustrate the occurrence of PNS adverse events after the administration of mRNA vaccines against COVID-19.

## 2. Materials and Methods

This study was conducted based on the guidance of the Cochrane Handbook for Systematic Reviews of Interventions [18] and adhered to the Preferred Reporting Items for Systematic Reviews and Meta-Analyses statement [19]. The study protocol has been registered at PROSPERO (CRD42022315956) before the authors completed this work.

### 2.1. Search Strategy

Two researchers (LYH and CHY) systematically searched for literature on neurological outcomes after the administration of mRNA COVID-19 vaccines in the Cochrane Library, MEDLINE, Embase, Scopus, and Web of Science before February 2022. Post-authorization safety surveillance reports were also searched. The following keywords with appropriate medical subject headings (MeSH terms in PubMed and Emtree in Embase) were applied: “peripheral nerve palsy,” “Bell’s palsy,” “mononeuropathy,” “Guillain–Barré syndrome,” “cranial nerve palsy,” or “Miller Fisher syndrome” and “Moderna,” “Spikevax,” “mRNA-1273,” “BNT162b2,” or “Pfizer vaccine.” No filters were used to restrict the language, publication date, age, sex, or article type. Furthermore, the two researchers searched a clinical trial registry for unpublished studies. The final search was done for references in the database before October 2022. The titles and abstracts of the studies were screened for the exclusion of irrelevant references, and the full text of potentially eligible studies were further reviewed by the two independent researchers. The reference lists of the studies identified to be eligible at the initial stage were also browsed for additional articles. Any disagreements between the two researchers were resolved by discussion with other senior authors.

### 2.2. Inclusion and Exclusion Criteria

The target designs for this study were randomized controlled trials (RCTs) and cohort studies, and those studies should have reported PNS adverse events after the administration of any SARS-CoV-2 vaccine with either a control group (with other COVID-19 vaccines or placebo) or unvaccinated cohort. This study was designed to examine the occurrence of PNS adverse events after the administration of mRNA COVID-19 vaccines, and large-scale studies are recommended for conducting meta-analyses of safety [18]. Therefore, studies with a total participant number fewer than 10,000 were excluded. In addition to the sample size, the exclusion criteria were as follows: non-comparative studies, studies only comparing patients with COVID-19 with vaccinated participants, and studies comparing other types of COVID-19 vaccines but not mRNA vaccines.

### 2.3. Data Collection (Outcomes)

Adverse events were defined mostly by diagnostic codes from the database or identified from the medical record review, contingent on the different designs of the included studies. Among all included studies, our goal was to analyze the risk of any new-onset PNS adverse events following the administration of mRNA vaccines against COVID-19 which were reported from large-scale studies; nevertheless, PNS adverse events in the included studies only covered Bell’s palsy and GBS. Other PNS adverse events including cranial polyneuritis, neuromuscular junction disorders, neuro-ophthalmological disorders, neurosensory hearing loss and dysautonomia were not analyzed in this study [20]. The number of adverse events and sample sizes were collected for each group (the mRNA COVID-19 vaccination and control groups). Data were extracted from the content or table or from the appendix and supplemental data of the included studies, and data presented as percentages were calculated in an adequate calculable form.

### 2.4. Quality Assessment

The Newcastle–Ottawa Scale (NOS) was used to evaluate the quality of the included studies. A star system was used to quantitatively assess three dimensions—selection, comparability, and outcome for cohort studies—with four, two, and three items, respectively. The number of stars ranged from zero to nine (7–9 stars, high-quality; 4–6 stars, medium-quality; <4 stars, poor-quality). The Cochrane Risk-of-Bias Tool was used to evaluate the quality of the included RCTs. Each domain was categorized as “low,” “high,” or “unclear” risk of bias.

### 2.5. Statistical Methods

All statistical analyses were performed using ReviewManager (version 5.4). The scale for the incidence rate of PNS adverse events was per 100,000 cases and pooled in a random-effects model because data were collected from various populations around the world. The associations between vaccination and the risk of PNS adverse events were estimated using relative risk. However, if zero-cell or very few cases were reported in the included studies, the Peto odds ratio (POR) was used. Point estimates were reported with 95% confidence intervals (CIs). Heterogeneity among the included studies was examined based on I^2^ statistics. When the I^2^ value was higher than 50%, the pooled estimate was highly heterogeneous across the included studies. Therefore, the results should be interpreted and implicated into clinical practice cautiously. For clinical implication, the Number Needed to Treat (NNT) was further calculated for significant findings based on inversed risk difference. Subgroup analyses were performed if data on the outcomes of interest were reported for each vaccination shot, type of mRNA COVID-19 vaccine, or age-group in the included studies.

## 3. Results

### 3.1. Characteristics of the Included Studies

According to the research strategy, 573 references were identified from the databases and registry platform. The other three references were found manually. After the exclusion of 67 duplicates and 431 irrelevant references, the full texts of 78 references were further retrieved. Fifty references were non-large-scale studies, which were subsequently excluded from this meta-analysis. Although the remaining 11 references were large-scale studies, they were still ongoing. Therefore, 17 references were included in this systematic review. After reviewing the full text, two studies did not provide sufficient data and were not put into quantitative synthesis [11,21]. Two BNT162b2 [5,22] and two mRNA-1273 [23,24] studies reported efficacy and safety issues among the same participants. In addition, the study by McMurry et al. exhibited an abnormal incidence rate of the outcome of interest [25] after we performed a sensitivity analysis. This meta-analysis was mainly based on results without data from McMurry et al.’s study. The results including data from the study by McMurry et al. are provided in the supplement. Finally, the total sample size was more than 33,000,000 cases from 13 observational comparative studies and two RCTs [11,17,21,22,23,24,25,26,27,28,29,30,31,32,33,34]. Figure 1 shows the study selection process. The characteristics of the included studies are listed in Table 1. The PNS adverse events reported in the included studies were Bell’s palsy and GBS. After a careful review, no data on other PNS adverse events were available for further analysis in the included studies. Quality assessment of the included studies is shown in Appendix A). In the two RCTs, risk of bias due to allocation, performance, attrition, and detection may be low, but other sources of bias might be at high risk. Regarding observational studies, all of the 13 studies may well control the biases from selection and outcome measurement, but only six of them did control for comparability.

### 3.2. Bell’s Palsy

Bell’s palsy was mentioned as a post-vaccination adverse effect of BNT162b2 or mRNA-1273 in eight studies [17,22,24,26,30,31,32,34]. The incidence rate of Bell’s palsy was about 6.07 per 100,000 persons (95% CI, 3.49–8.65) after the first dose of mRNA vaccination, and decreased to 5.06 per 100,000 persons (95% CI, 1.97–8.16) after the second dose of mRNA vaccination. While the study by McMurry et al. was removed from the pooled analysis, the incidence rates of post-vaccinated Bell’s palsy after the first and the second dose of mRNA vaccination were 5.30 and 4.40 per 100,000 cases, respectively (Appendix A). However, without specific information on the dose of mRNA vaccination, the incidence rate of post-vaccinated Bell’s palsy reached 31.08 per 100,000 persons (95% CI, 1.44–60.71) in those studies (Appendix A). Compared to unvaccinated groups, individuals who received BNT162b2 or mRNA-1273 had a marginal increase in the occurrence of Bell’s palsy based on 2,400,981 cases (POR, 1.36; 95% CI, 1.03–1.79), and the results of the five studies revealed a moderate heterogeneity (I^2^, 37%) (Figure 2A). The findings might not achieve clinical significance because of an extremely high NNT of 36,550. When McMurry et al.’s study is included, heterogeneity increased significantly (I^2^, 82%) (Appendix A).

Due to limited information, further analysis of Bell’s palsy could only be separated by shots of BNT162b2, mRNA-1273 SARS-CoV-2 Vaccine, and a comparison of the two mRNA vaccines. Four studies provided data comparing the occurrence of adverse events after the first or second dose of BNT162b2 with that in the unvaccinated group [17,22,26,32]. Data included the first dose (*n* = 520,818), the second dose (*n* = 316,566), and both doses (*n* = 2,368,202). Three studies had separate data of the different doses of BNT162b2 [17,26,32]. Individuals who received the first shot of BNT162b2 did not have a significantly higher risk of Bell’s palsy than the unvaccinated group (POR, 1.05; 95% CI: 0.60–1.85; I^2^, 31%) (Figure 2B). Three studies reported the occurrence of Bell’s palsy after the second shot of BNT162b2, while one of them did not show the occurrence of Bell’s palsy in both the BNT162b2 and unvaccinated groups. The pooled results showed a similar risk of Bell’s palsy between individuals who received the second shot of BNT162b2 and the unvaccinated group (POR, 0.92; 95% CI: 0.40–2.08); however, the pooled results were highly heterogeneous (I^2^, 73%). One study did not report the occurrence of Bell’s palsy after each BNT162b2 shot [22]; therefore, this study also synthesized the total incidence of Bell’s palsy after BNT162b2 without separation according to vaccine doses. The pooled results showed no significant difference in the risk of Bell’s palsy between the BNT162b2 and unvaccinated groups (POR, 1.23; 95% CI, 0.94–1.62); however, moderate to high heterogeneity existed across the included studies. The pooled results were not seriously biased by data from the study by McMurry et al. (Appendix A).

Three studies reported the occurrence of Bell’s palsy after the administration of the mRNA-1273 SARS-CoV-2 vaccine (*n* = 43,851) [24,25,26]. One study noted no Bell’s palsy event after vaccination. Based on these data, except for the study by McMurry et al. [25], the risk of Bell’s palsy did not significantly differ between individuals who received mRNA-1273 SARS-CoV-2 vaccination and the unvaccinated (POR, 2.66; 95% CI, 0.71–10.04) (Appendix A). However, these two studies did not provide data according to vaccine doses. This result was not seriously affected if data from the study by McMurry et al. were included (Appendix A).

With regard to the comparison of Bell’s palsy after BNT162b2 and mRNA-1273, data can be extracted from four studies (*n* = 10,937,523) [26,30,31,34], and the pooled result showed that risk of Bell’s palsy after BNT162b2 was significantly higher than after mRNA-1273 (POR, 1.64; 95% CI, 1.41–1.90; I^2^, 0%) if excluding the data from the study by McMurry et al. (Figure 2B). These results were also not seriously biased if the data from the study by McMurry et al. was included (Appendix A) [25].

### 3.3. Guillain-Barré Syndrome

Five large-scale studies reported the occurrence of GBS after the administration of BNT162b2 and mRNA-1273 vaccinations [17,27,29,30,34]. The incidence rate of GBS after mRNA vaccination was 0.19 per person (95% CI, 0.13–0.25), but the incidence rate of GBS would increase to 1.3 per 100,000 persons based on doses of mRNA vaccination (95% CI, 0.42–2.18) with extremely high heterogeneity (I^2^, 99%) (Appendix A). Due to the limited data, we only compared the occurrence of GBS of BNT162b2 vs. unvaccinated and BNT162b2 vs. mRNA-1273. A large effect size in the comparison of GBS between the BNT162b2 and unvaccinated groups was noticed, which raised concerns based on statistical perspectives (POR, 7.39; 95% CI, 0.15–372.38), although the finding was non-significant. The risk of GBS of BNT162b2 was significantly higher than that of mRNA-1273 (POR, 2.85; 95% CI, 1.61–5.04) with low heterogeneity (I^2^, 0%) (Figure 3).

## 4. Discussion

This meta-analysis evaluated the occurrence of PNS adverse events including Bell’s palsy and GBS after the administration of BNT162b2 or mRNA-1273 vaccination by analyzing large-scale studies. The overall incidence of Bell’s palsy after the administration of the mRNA vaccine was marginally increased compared with the unvaccinated group. Regarding individual mRNA vaccines, neither BNT162b2 nor mRNA-1273 increased the risk of Bell’s palsy; however, BNT162b2 introduced more post-vaccinated Bell’s palsy than mRNA-1273. Considering GBS, the risk after BNT162b2 was similar to the unvaccinated group, but higher than individuals receiving mRNA-1273.

Bell’s palsy is the most frequent acute mononeuropathy and results in the partial or complete inability to automatically move the affected side of the facial muscles [35]. Whether mRNA vaccines against SARS-CoV-2 would increase risk of Bell’s palsy were controversial from observational studies. Two studies disclosed an increased risk of Bell’s palsy after BNT162b2 and mRNA-1273 vaccinations [11,36], while the other two studies underpinned by safety surveillance databases found no association [16,21]. This large scale meta-analysis revealed a marginal increase of incidence of Bell’s palsy after mRNA vaccinations compared to the unvaccinated groups, and this result suggested that post-vaccination immune reactions possibly affect peripheral nerves [15]. A possible explanation is the secretion of inflammatory cytokines, particularly type 1 interferon, after mRNA vaccination [37]. Type 1 interferons play an important role in mRNA vaccine-induced immune responses, including dendritic cell modulation and T-cell differentiation, and possibly in the development of autoimmune responses against myelin sheaths [37]. Consistent with this hypothesis, in patients with Bell’s palsy, the serum levels of interferons increased [38]. Additional evidence is provided by a case series of patients with hepatitis C receiving interferon-α treatment who were more vulnerable to developing Bell’s palsy [39,40,41], suggesting that the use of interferon therapy affects myelin sheaths and leads to neuropathy [39].

The background incidence rates of Bell’s palsy in different databases were reported to be between 6.21 and 238.73 per 100,000 persons [42]. In this study, the incidence of Bell’s palsy after the administration of BNT162b2 or mRNA-1273 vaccine in this study was 31.08 per 100,000 persons, and the data is lower than background incidence rates of most databases. Although this meta-analysis revealed a marginal increase in the incidence of Bell’s palsy after vaccination, the overall occurrence of Bell’s palsy during the COVID-19 era is decreasing. The pathogenesis of Bell’s palsy has been postulated as a post-infectious inflammatory process of the affected facial nerve because of significantly high cytokine levels [43]. During the COVID-19 era, community-acquired respiratory infections significantly decreased because of lockdown, the maintenance of social distancing, and mask-wearing [44,45]. The reduction of acute infections is possibly associated with the overall decreased incidence of Bell’s palsy. Contrarily, after SARS-CoV-2 infection, the incidence of post-infectious Bell’s palsy increased to 82 per 100,000 persons [46], and the significant risk of Bell’s palsy after SARS-CoV-2 infection further warranted the requirement of vaccine prophylaxis. The main treatment for Bell’s palsy is oral corticosteroids to reduce inflammation and improve facial nerve recovery [47]. The prognosis for idiopathic Bell’s palsy is good even without treatment, and patients usually recover within 6 months [48]. However, the prognosis for Bell’s palsy induced by mRNA vaccination or direct SARS-CoV-2 infection is unclear. In an Israeli population-based study, the attributable risk of post-vaccinated Bell’s palsy was found to be higher in participants older than 65 years and those who have had Bell’s palsy previously [49]. Therefore, long-term monitoring is required after mRNA vaccinations, particularly in older individuals and those with a history of Bell’s palsy.

GBS is an immune-mediated polyradiculopathy which results in acute flaccid paralysis. The acute onset of neurological symptoms is usually preceded by an infective illness, followed by progressive weakness reaching plateau in 4 weeks [50] Effective treatments include intravenous immunoglobulin and plasma exchange. Up to 40% of patients do not improve in the first 4 weeks following treatment; however, most patients show extensive recovery in the first year after disease onset [51]. Although GBS has been recognized as an adverse event after ChAdOx1-nCoV-19 vaccination and would result in significant morbidity [52], data on the occurrence of GBS after the administration of mRNA vaccines against COVID-19 are limited. Despite the lack of head-on-head comparison between individuals who received mRNA vaccines and unvaccinated people, in this study the overall incidence of GBS after the administration of BNT162b2 or mRNA-1273 vaccine was 0.27 per 100,000 persons, which is lower than the historical worldwide incidence of 0.8–1.9 cases per 100,000 persons [53]. In contrast, SARS-CoV-2 natural infection might cause GBS with a prevalence rate of 15 cases per 100,000 persons [54], and the significantly higher risk further consolidates the requirement of vaccine prophylaxis. Unlike *C. jejuni* or Cytomegalovirus infection, which causally induces the production of antibodies against myelin-associated antigens, such as ganglioside, no direct evidence of molecular mimicry or coronavirus invasion underlying GBS manifestation after SARS-CoV-2 infection has been reported [55]. The causality and pathogenesis of GBS related to SARS-CoV-2 infection, as well as the clinical course and prognosis of this specific group of patients, warrants comprehensive studies. Interestingly, regarding post-vaccinated Bell’s palsy and GBS, BNT162b2 revealed higher risks than mRNA-1273 in this study. Higher post-vaccinated side effects of BNT162b2 were also observed in a cohort study of US veterans [56]. Although both mRNA vaccines encoded full-length spike proteins of SARS-CoV-2, each uses different delivery systems, mRNA amounts and dosing schedules [56]. Therefore, the antibody responses to these two mRNA vaccines were different [57]; however, the exact mechanisms for different post-vaccinated side effects in these two mRNA vaccines are unclear and warrant further investigation.

This study has some limitations. First, our synthesis is based on limited numbers of RCTs, and most evidence comes from observational studies. Observational studies have their own unavoidable bias in finding causation due to complicated confounding and predisposing factors. However, relatively small-scale studies are inappropriate for this topic due to the naturally low incidence rate of PNS diseases. If we included those relatively small-scale studies, the odds ratio of PNS adverse events after mRNA vaccination may be seriously underestimated, and that is why large-scale studies are recommended for synthesis on the safety issue. In addition to the prevention of underestimated risk in the present design, the pooled findings are still reliable due to the very large sample size (*n* > 33,000,000), although the eligibility criteria result in small numbers of studies for synthesis. Second, our main findings and conclusion are based on analysis without data from the study by McMurry et al. because of its abnormal prevalence rate in the unvaccinated group [25]. In that study, for instance, the incidence of Bell’s palsy in cases without COVID-19 vaccination was approximately 100 per 100,000 persons, which is higher than the common situation [58]. We noticed that the high background incidence rate could cause some biases in the pooled estimates. Third, as we mentioned above, only six observational studies in the present synthesis are adequate in comparability, which may threaten the confidence of the findings. Anyone applying our findings should be aware of this potential bias introduced by unclear comparability. Fourth, the pooled estimates of GBS in this study might have been affected by selection bias and potential risk in incomplete follow-up. Regarding selection bias, two of the five studies mainly included a high-risk population [13,30]. In contrast, one of these studies only reported percentages of PNS adverse events rather than the number of them [30]; therefore, the outcome was estimated from the percentage and numbers of samples. The data conversion might have resulted in the certainty of the pooled results. Third, the PNS adverse events we analyzed only included Bell’s palsy and GBS, and available data on other PNS adverse events were insufficient for analysis.

## 5. Conclusions

To date, more than 485 million individuals have been affected by COVID-19, and more than 4.5 billion individuals have been fully vaccinated. Currently, conducting a prospective study to analyze the correlation between COVID-19 vaccination and neurological adverse events is difficult. Although this study found a marginal increase in the incidence of Bell’s palsy after BNT162b2 or mRNA-1273 vaccination compared with the unvaccinated group, these vaccines are mandatory to prevent severe SARS-CoV-2 infection and its consequences. With the opportunity of vaccination campaigns on such a large scale, further investigation and surveillance of post-vaccination neurological adverse events should also be established.

## Figures and Tables

**Figure 1 vaccines-10-02174-f001:**
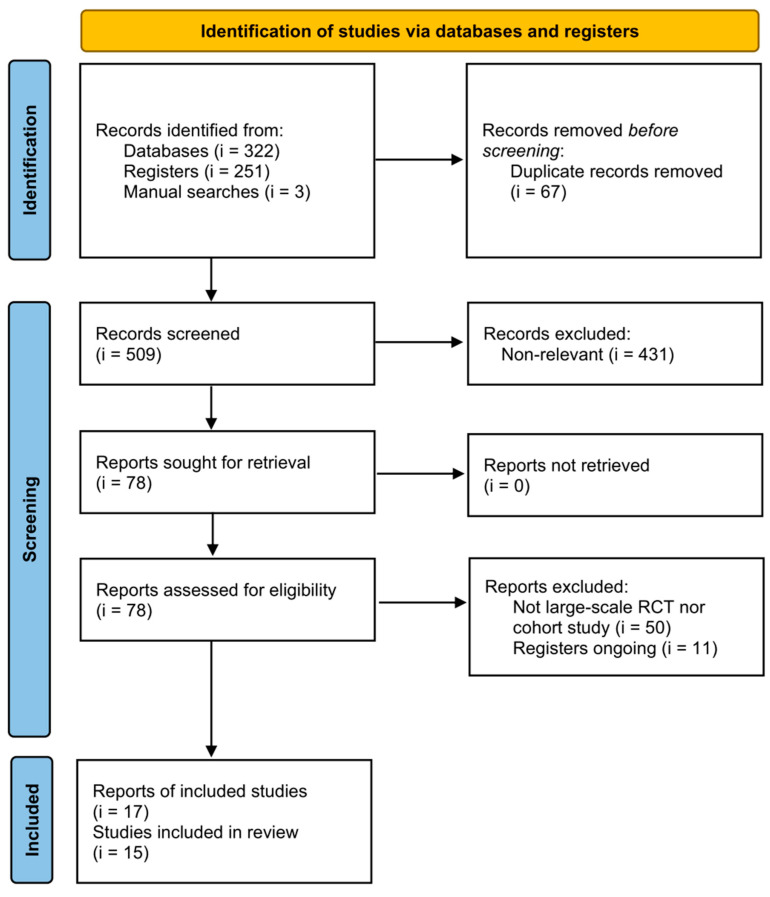
Preferred Reporting Items for Systematic Review and Meta-Analysis flow diagram of evidence selection.

**Figure 2 vaccines-10-02174-f002:**
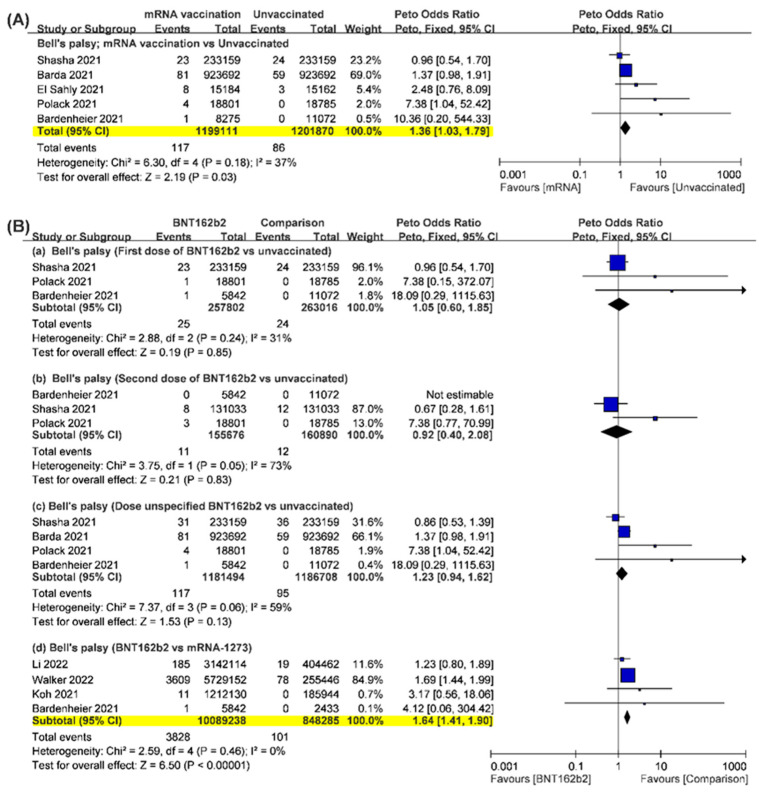
Forest plot analysis of the incidence of Bell’s palsy after the administrations of mRNA vaccines versus the unvaccinated group (**A**) and BNT162b2 vaccines versus other comparisons (**B**). Statistical significance is highlighted in yellow.

**Figure 3 vaccines-10-02174-f003:**
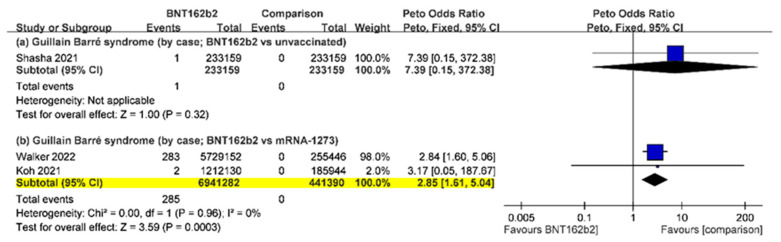
Forest plot analysis of the incidence of Guillain–Barré syndrome after the administration of BNT162b2 versus other comparison. Statistical significance is highlighted in yellow.

**Table 1 vaccines-10-02174-t001:** Characteristics of the studies in this systematic review and meta-analysis.

Trial/Author	Location	Period	Treatment	Dose	Sex (Male/Female)	Mean Age (Years)
NCT04470427	United States	2020/07/27 to	mRNA-M	1, 2 (but mixed)	7917/7263	51.4
(Baden, El Sahly)		2020/10/23	Unvaccinated	Nil	8057/7109	51.3
NCT04368728	152 sites	2020/07/27 to	mRNA-B	1, 2 (but mixed)	9639/9221	52
(Polack, Stephen)	worldwide	2020/11/14	Unvaccinated	Nil	9436/9410	52
Barda et al.	Israel	2020/12/20 to	mRNA-B	1, 2 (but mixed)	461,590/423,238	38
		2021/05/24	Unvaccinated	Nil	461,590/423,238	38
Bardenheier et al.	United States	2020/12/18 to	mRNA-B/M	1, 2 (separated)	1st 3191/5301, 2nd 3072/5161	>65 (without mean age)
		2021/03/07	Unvaccinated	1, 2 (separated)	4128/6731	>65 (without mean age)
Garcia-Grimshaw et al.	Mexico	2020/12/24 to	mRNA-B	Non-specific	All (dose): 16,646,623	>18 (without mean age)
		2021/10/29	mRNA-M	Non-specific	All (dose): 2318,057	>18 (without mean age)
Hanson et al.	United States	2020/12/13 to	mRNA-B	1, 2 (but mixed)	Overall: 14,637,020	>12 (without mean age)
		2021/11/13	mRNA-M	1, 2 (but mixed)		>12 (without mean age)
Keh et al.	England	2021/01/01	mRNA-B	Non-specific	Overall (national database) > 10,000 but unclear	NR
		2021/11/07	mRNA-M	Non-specific		NR
Koh et al.	Singapore	2020/12/30 to	mRNA-B	1, 2 (but mixed)	Overall (estimated from percentage and total cases):	Overall:
		2021/04/20	mRNA-M	1, 2 (but mixed)	761,950/636,124	59
Li (Xintong) et al.	United Kingdom	2020/09/01	mRNA-B	1, 2 (separated)	Overall: 8330,497	>18 (without mean age)
	and Spain	2021/06/23	mRNA-M	1, 2 (separated)		>18 (without mean age)
McMurry et al.	United States	2020/12/01 to	mRNA-B	1, 2 (separated)	20,695/31,099	53.83
		2021/04/20	mRNA-M	1, 2 (separated)	7713/8758	63
			Unvaccinated	Nil	28,408/39,857	55.61
Renoud et al.	Worldwide	Before	mRNA-B	1, 2 (but mixed)	Overall >10,000 but unclear	NR
		2021/03/09	mRNA-M	1, 2 (but mixed)		NR
Rosenblum et al.	United States	2020/12/14 to	mRNA-B	1, 2 (but mixed)	45,157/116,587 (other 2925 were unknown)	Unclear
		2021/06/14	mRNA-M	1, 2 (but mixed)	43,140/129,475 (other 3201 were unknown)	Unclear
Sato et al.	United States	Before	mRNA-B	1, 2 (but mixed)	Overall >10,000 but unclear	NR
		2021/04/30	mRNA-M	1, 2 (but mixed)		NR
Shasha et al.	Israel	2020/12/19 to	mRNA-B	1, 2 (separated)	1st 114,634/118,525; 2nd 64,580/66,453	Unclear
		2021/02/12	Unvaccinated	Nil	1st 121,453/111,706; 2nd 64,580/66,453	Unclear
Walker et al.	England	2020/07/01 to	mRNA-B	1, 2 separately	All: 5,729,152	>18 (without mean age)
		2021/07/07	mRNA-M	1, 2 separately	All: 255,446	>18 (without mean age)

mRNA-B, BNT162b2 vaccine; mRNA-M, mRNA-1273 vaccine; NR, no report.

## Data Availability

Data is contained within the article.

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
