# Peer review of "Peripheral Nervous System Adverse Events after the Administration of mRNA Vaccines: A Systematic Review and Meta-Analysis of Large-Scale Studies"

_vaccines, 2022, doi:10.3390/vaccines10122174_

Round 1

Reviewer 1 Report

This review and meta-analysis of the risk of peripheral nervous system (PNS) adverse events after the administration of mRNA vaccines against COVID-19 follows standard protocols for such studies. Overall, the paper is well organized, well written, and the analyses appear to be well done. 

Here are a couple of items to attend to in a revision:

First, in order to facilitate accessibility of the paper's contents to readers of Vaccines who may not be as well informed as the authors about the intricacies of the PNS adverse events that are the focus of the study, it would be good to include an Appendix that gives brief definitions and descriptions of Bell’s palsy and the Guillain–Barré syndrome (symptoms, causes, etc.). 

Second, in the Abstract and Conclusion sections of the paper, it would be good to give a general numeric indication of the orders of magnitude of the "rarity" and "higher risk". These are very small, but, again, some indication thereof will be useful information for readers.

Reviewer 2 Report

The manuscript represents an interesting analysis concerning the unwanted effects of mRNA vaccine. However, several issues must be addressed before considering the manuscript for publication.

-The first issue is related with the low number of trials, some of them difficult to combine based on the approaches and analysis.

-The second issue refers to a bias found in the first figure of the supplemental material, which affects the table and conclusions. The rationale for analyzing the effect of each dose but the manuscript of McMurry differs from the others and should be carefully screened. 

-The conclusions are overstated based on the low number of manuscripts analysed. 

-The limitations of the study are several. and should be stated. 

Round 2

Reviewer 2 Report

The authors have made the requested changes in the manuscript. It is now suitable for publication